# Zearalenone Induces Apoptosis in Porcine Endometrial Stromal Cells through JNK Signaling Pathway Based on Endoplasmic Reticulum Stress

**DOI:** 10.3390/toxins14110758

**Published:** 2022-11-03

**Authors:** Jie Zhao, Sirao Hai, Jiawen Chen, Li Ma, Sajid Ur Rahman, Chang Zhao, Shibin Feng, Yu Li, Jinjie Wu, Xichun Wang

**Affiliations:** 1College of Animal Science and Technology, Anhui Agricultural University, Hefei 230036, China; 2Department of Food Science and Engineering, School of Agriculture and Biology, Shanghai Jiao Tong University, Shanghai 200240, China; 3Anhui Province Engineering Laboratory for Animal Food Quality and Bio-Safety, Hefei 230036, China

**Keywords:** zearalenone, porcine endometrial stromal cells, JNK signaling pathway, endoplasmic reticulum stress

## Abstract

Zearalenone (ZEA) is an estrogen-like mycotoxin characterized mainly by reproductive toxicity, to which pigs are particularly sensitive. The aim of this study was to investigate the molecular mechanism of ZEA-induced apoptosis in porcine endometrial stromal cells (ESCs) by activating the JNK signaling pathway through endoplasmic reticulum stress (ERS). In this study, ESCs were exposed to ZEA, with the ERS inhibitor sodium 4-Phenylbutyrate (4-PBA) as a reference. The results showed that ZEA could damage cell structures, induce endoplasmic reticulum swelling and fragmentation, and decreased the ratio of live cells to dead cells significantly. In addition, ZEA could increase reactive oxygen species and Ca^2+^ levels; upregulate the expression of GRP78, CHOP, PERK, ASK1 and JNK; activate JNK phosphorylation and its high expression in the nucleus; upregulate the expression Caspase 3 and Caspase 9; and increase the Bax/Bcl-2 ratio, resulting in increased apoptosis. After 3 h of 4-PBA-pretreatment, ZEA was added for mixed culture, which showed that the inhibition of ERS could reduce the cytotoxicity of ZEA toward ESCs. Compared with the ZEA group, ERS inhibition increased cell viability; downregulated the expression of GRP78, CHOP, PERK, ASK1 and JNK; and decreased the nuclear level of p-JNK. The Bax/Bcl-2 ratio and the expression of Caspase 3 and Caspase 9 were downregulated, significantly alleviating apoptosis. These results demonstrate that ZEA can alter the morphology of ESCs, destroy their ultrastructure, and activate the JNK signaling via the ERS pathway, leading to apoptosis.

## 1. Introduction

Mycotoxins are a class of toxic secondary metabolites produced during the growth of molds. They are among the most potent natural toxic substances, causing significant pollution of cereal crops worldwide [1,2]. Zearalenone (ZEA) is a non-steroidal mycotoxin widely found in cereals. Numerous in vivo and in vitro studies have shown that the dominating biological activity of ZEA is its estrogenic activity, when it binds to estrogen receptors (ERs), it can induce estrogen-like response [3]. According to the relative intensity factors (RPFs), it can be found that α-ZEL is the most influential factors among ZEA and its metabolites, and the toxicity ranking is as follows: α-ZEL > α-ZAL > ZEA ≈ ZAN ≈ β-ZAL > β-ZEL. A large number of data statistics show that in all types of pigs, the production of α-ZEL greatly exceeds the production of β-ZEL and other reducing metabolites, and the overall toxicity is the highest after ZEA is completely hydrolyzed [3]. ZEA and its metabolites are most sensitive to estrogenic activity in pigs. The Joint Food and Agriculture Organization of the United Nations (FAO) and the World Health Organization (WHO) Expert Committee on Food Additives (JECFA) evaluated ZEA and the provisional maximum tolerated daily intake (PMTDI) of 0.25 μg/kg b.w. was determined [4]. Studies have shown that in the last 10 years, the positive detection rate of ZEA in cereals was about 46%, and the maximum level was 3049 μg/kg [5]. ZEA has potent estrogenic activity that can compete with estrogen to bind to estrogen receptors, which disrupts the endocrine system leading to reproductive problems in various animals, among which pigs are the most susceptible [6,7]. When female animals were poisoned by ZEA, their reproductive capacity decreased, resulting in follicular atresia, ovarian atrophy, uterine wall multi-layer cell hypertrophy, tissue congestion, and edema. This led to decreased fertility, spontaneous abortion, stillbirth, and deformed fetuses, which could seriously affect the economic development of animal husbandry [8]. Studies have shown that doses of ZEA up to 1000 μg/kg can lead to hypertensive syndrome in pigs; whereas, ZEA causes sterility in sows by inciting a malfunction of the ovary and other reproductive system disorders [9].

Endometrial stromal cells (ESCs) are spindle-shaped or stellate-shaped cells with a low degree of differentiation in the endometrial stromal layer. ESCs produce a large number of growth factors and cytokines that affect epithelial function, such as epidermal growth factor (EGF), insulin-like growth factor 1 (IGF-1), and hepatocyte growth factor (HGF). When exposed to toxins, they change their morphology and function, which can be used as a basis for studying the molecular mechanism of endometrial cell differentiation and metabolism [10,11].

The endoplasmic reticulum (ER) is a central organelle in a number of important biological processes, which functions to maintain the stability of the intracellular environment [12]. Normally, the protein folding ability of ER is in dynamic balance with the body’s protein synthesis ability, when ischemia, hypoxia, injury, and other deleterious events occur, the ER microenvironment changes, resulting in endoplasmic reticulum stress (ERS) [13,14]. Moderate ERS restores homeostasis by reducing protein translation and promoting chaperone production [15]. While ERS persists, protein kinase RNA-like ER kinase (PERK) and inositol requiring enzyme-1 *(*IRE1*)* can be separated from glucose-regulated protein 78 (GRP78), then induce PERK and IRE 1 autophosphorylation, the subsequent activation of downstream activated transcription factor 4 (ATF4)-C/EBP-homologous protein (CHOP) and apoptosis signal-regulated kinase 1 (ASK1)-c-Jun N-terminal kinase (JNK) signaling pathways leads to continuously high expression of CHOP and JNK, which promote apoptosis [10,16]. ASK1 can be activated by a variety of pathological stimuli, and activated ASK1 in turn activates downstream kinases such as JNK and p38, leading to apoptosis [17,18]. JNK is another subclass of mitogen-activated protein kinase (MAPK), following stimulation, JNK accumulates rapidly and significantly in the nucleus, leading to changes in apoptosis-related gene expression. Studies have shown that transient JNK activation has both pro- and anti-apoptotic effects, while sustained JNK activation promotes apoptosis [19,20,21]. Therefore, this experiment hypothesized that ZEA could induce ERS and activate the JNK pathway, thereby leading to apoptosis.

In this study, different concentrations of ZEA were added into the culture medium of ESCs and the ERS inhibitor Sodium 4-Phenylbutyrate (4-PBA) was added as a control. Cell counting kit-8 (CCK-8) assays, transmission electron microscopy, flow cytometry, quantitative real-time reverse transcription PCR (qRT-PCR), and western blotting were used to investigate the effects of ZEA on porcine ESCs. We aimed to clarify the molecular mechanism of ERS activating JNK signaling pathway in ZEA-induced toxic injury of porcine ESCs.

## 2. Results

### 2.1. Effects of ZEA on Cell Viability

As shown in Figure 1, the viability of ESCs was significantly decreased compared with that in the control group (ZEA concentration of 0 μM) at a ZEA concentration of 5 μM (*p* < 0.01), and cell viability decreased with increasing ZEA concentration in a dose-dependent manner. When the ZEA concentration reached 15 μM, cell viability was close to 50% (IC50 = 15 μM).

### 2.2. Effects of ZEA on ESC Growth

In Figure 2A, green fluorescence represents live cells and red fluorescence represents dead cells. In the control group, the green fluorescence was strong and evenly distributed. With increasing ZEA concentration, the red fluorescence gradually increased, indicating a change the cell distribution toward dead cells, with a corresponding significant decrease in live cells. These results suggested that ZEA could affect the growth of ESCs.

### 2.3. Effect of ZEA on the Cell Ultrastructure

Figure 2B shows that the cell morphology of the control group was complete, with normal intracytoplasmic organelles, mitochondria, and a complete ER. Nuclear chromatin distribution was uniform and the nuclear membrane was clear. With increasing ZEA concentration, the cells shrank and the organelles ruptured. When the concentration reached 15 μM, significant cell shrinkage was observed, the cell membrane was damaged, the number of autophagy vesicles increased, the ER and mitochondrial cristae were ruptured, and the structure was unclear. The edges of the nuclear membrane were observed to be attached to the nuclear chromatin, which was highly concentrated.

### 2.4. Effect of ZEA on Ca^2+^ Levels, ROS Levels, and the Apoptosis Rate

Figure 3A,C show that the Ca^2+^ level was significantly higher in the ZEA group than in the control (CON) group (*p* < 0.01), but was not significantly different between group treated with the endoplasmic reticulum inhibitor 4-PBA and the CON group (*p* > 0.05). The Ca^2+^ level was significantly increased in the 4-PBA + ZERA group compared with that in the CON group (*p* < 0.01), but was significantly decreased compared with that in the ZEA group (*p* < 0.01).

As shown in Figure 3B,D, compared with that in CON group, the ROS level in the 4-PBA group was not significantly different (*p* > 0.05), whereas the ROS level in the ZEA group was significantly increased (*p* < 0.01). The level of ROS in the P + Z group was significantly decreased compared with that in the ZEA group (*p* < 0.01).

Figure 3E,F show that the apoptosis rate in the control group was only 1.34%, whereas the apoptosis rate in the ZEA group was significantly higher at 14.4% (*p* < 0.01). There was no significant difference between the apoptosis rate in the 4-PBA group and the CON group (*p* > 0.01). The apoptosis rate in the P + Z group decreased significantly compared with that in the ZEA group.

### 2.5. Effects of ZEA on Gene Expression in Porcine ESCs

4-PBA was added to the ESCs to explore the molecular mechanism of apoptosis induction by ZEA through the ERS pathway. As shown in Figure 4, the relative expression level of ERS marker genes, apoptosis-related genes, and JNK were detected. Compared with those in the CON group, the relative expression levels of CHOP, GRP78, HSP70 (encoding heat shock protein 70), PERK (encoding protein kinase RNA-like ER kinase) and ASK1 mRNA in the ZEA group were significantly increased (*p* < 0.01). Moreover, the relative expression of JNK mRNA and apoptosis-related genes Caspase 3 and Caspase 9, and the Bax/Bcl-2 ratio were significantly increased (*p* < 0.01). However, the 4-PBA group showed no significant changes in the expression levels of these genes compared with those in the CON group (*p* > 0.05). Compared with those in the ZEA group, the ZEA-induced changes in ERS-related and apoptosis-related genes were significantly alleviated in the P + Z group (*p* < 0.05 or *p* < 0.01), and JNK expression and the Bax/Bcl-2 ratio decreased significantly (*p* < 0.01).

### 2.6. Effects of ZEA on Protein Levels in Porcine ESCs

#### 2.6.1. Levels of ERS-Related Proteins

As shown in Figure 5, compared with those in the CON group, the levels of ERS-related marker proteins GRP78, CHOP, PERK, and ASK1 were significantly increased by ZEA treatment (*p* < 0.01), while none of the proteins showed a significant change in 4-PBA group compared with that in the CON group (*p* > 0.05). After adding the ERS inhibitor 4-PBA and ZEA, the levels of ASK1, GRP78, CHOP, and PERK in the P + Z group were significantly decreased compared with those in the ZEA group (*p* < 0.01). The results showed that ZEA can induced ERS in ESCs, and when ERS persisted, it might cause cell damage (Figure 5A,E–H).

#### 2.6.2. Expression of JNK

Figure 5B,I, show that compared with that in the CON group, ZEA increased the intracellular level of p-JNK, and significantly increased the p-JNK/JNK ratio (*p* < 0.01). After the addition of the ERS inhibitor 4-PBA, the p-JNK/JNK ratio in the P + Z group was significantly lower than that in the ZEA group (*p* < 0.01). 

#### 2.6.3. Nuclear Level of p-JNK

Figure 5D indicates the effect of ZEA on the level of p-JNK in the nucleus of porcine ESCs. As shown in Figure 5M, the levels of p-JNK in the nucleus of the ZEA group and P + Z group were significantly increased compared with those in the CON group (*p* < 0.01), while no significant difference was detected in the 4-PBA group (*p* > 0.05). The p-JNK level in the nucleus of P + Z group was significantly lower than that in the ZEA group (*p* < 0.01). The results showed that ZEA induced porcine ESCs ERS and activated phosphorylation of JNK, which subsequently showed a significantly higher level in the nucleus.

#### 2.6.4. Expression Levels of Apoptosis-Related Proteins

The effects of ZEA on apoptosis-related proteins in porcine ESCs are shown in Figure 5C. The results show that compared with that in the CON group, there was no significant difference in the Bax/Bcl-2 ratio, or the levels Caspase 3 and Caspase 9 protein in the 4-PBA group (*p* > 0.05); however, the Bax/Bcl-2 ratio and the level Caspase 3 (*p* < 0.01) and Caspase 9 (*p* < 0.05) were significantly increased in the ZEA group. Compared with those in the ZEA group, the Bax/Bcl-2 ratio, the Caspase 3 (*p* < 0.01), and the Caspase 9 level (*p* < 0.05) were significantly decreased in the P + Z group.

## 3. Discussion

ZEA can bind to the estrogen receptor; therefore, female pigs exposed to ZEA show different degrees of adverse reactions, mainly reflected in reproductive toxicity. In addition, pregnant sows are susceptible to the effects of ZEA [22]. Previous studies by our research group have shown that ZEA can induce oxidative stress in porcine Sertoli cells (SCs) and subsequently induces apoptosis, suggesting that ZEA has some reproductive toxicity in male pigs [3,4]. According to previous reports, sows fed diets containing 1.3 mg/kg ZEA experienced systemic toxicity and histopathological changes in the liver and kidney. Similarly, blood toxicity was found in mice exposed to ZEA, while also reducing total protein and albumin levels [23]. Previous studies have shown that ZEA is toxic to human and animal health; however, research has focused on the reproductive toxicity of ZEA in females, mainly targeting granulosa cells [24,25]. However, ZEA, as an estrogen-like toxin, has different effects on female and male animals. ESCs are present in the endometrial stromal layer, and can change their morphology and function when exposed to ZEA. Therefore, it is important to investigate the reproductive toxicity of ZEA to porcine ESCs.

Mycotoxins are the most important risk factors in the feed supply chain. Mycotoxins are mainly present in cereals, accounting for 50–70% of feed weight. Wheat, corn, barley, oats, and rye are the most susceptible to mycotoxin contamination, which causes adverse effects in humans and animals [26]. Studies have shown that the morphology of porcine hippocampal neurons, intestinal porcine epithelial cells (IPECs), and SCs becomes irregular when the cells were exposed to mycotoxins, and the cells then undergo atrophy and cell death. The ER and mitochondrial cristae are broken, with unclear structures [4,27]. The edge of the nuclear membrane becomes connected to the nuclear chromatin, which is highly concentrated. Previous studies showed that as the concentration of ZEA increased, the number of damaged cells increased [28,29]. In the present study, ESCs exposed to ZEA showed decreased survival in a ZRA concentration-dependent manner, and different degrees of damage were observed in the ultrastructure of cells, especially swelling and rupture of mitochondria and the ER. Chromatin aggregation, and nuclear membrane and cell membrane damage increased as the ZEA concentration increased. In this context, we suggest that the poisoning of sows ESCs by ZEA might first cause ERS, which then leads to apoptosis via a specific pathway, and showed a dose-dependent relationship. The results are consistent with previous studies.

Unfolded protein reaction (UPR) signaling involves IRE1, PERK, and activating transcription factor 6 (ATF6). UPR can maintain ER homeostasis at the transcriptional and translational levels, and reduces the synthesis of unfolded and resolvable proteins. ERS is a self-protection and defense mechanism of eukaryotic cells. However, if ERS persists, UPR is insufficient to maintain ER homeostasis. At the same time, the UPR would activate apoptosis-related factors to induce cell apoptosis [30,31]. PERK belongs to the serine/threonine protein kinase family. It is an important protein receptor that exists on the ER membrane and is involved in the adaptive response of cells. Normally, PERK binds to GRP78 in an inactive state. If ERS persists, the activated PERK can form an oligomer by autophosphorylation to activate the downstream ATF4-CHOP pathway [32]. GRP78 is an important marker protein of ERS. When ERS occurs, PERK and GRP78 are rapidly dissociated and activated by autophosphorylation of their cytoplasmic domains. Phosphorylated PERK activates the 5th serine of eukaryotic initiation factor 2a (elF2a), which upregulates ATF4 expression and further upregulates CHOP expression, ultimately activating downstream signal transduction and triggering apoptosis [10]. Similar to PERK, excessive ERS might lead to autophosphorylation of IRE1, which then initiates the downstream ASK1-JNK signaling pathway through its phosphorylation kinase activity. The results of the present study showed that the relative expression levels of ASK1, PERK, GRP78, and CHOP mRNAs increased significantly in ESCs exposed to ZEA compared with the control group, as did their protein levels. ZEA-induced ERS was significantly alleviated by the addition of the ERS inhibitor 4-PBA.

The dynamic balance of Ca^2+^ is the basis for maintaining cellular structure and function. The ER is an important calcium reservoir in cells [33]. When cells are in an ERS state, the calcium balance is disturbed, leading to destruction of the cell or organelle membrane, which ultimately induces apoptosis. Studies have shown that ZEA can trigger apoptosis and death through ERS-dependent signaling [34]. ROS play an important role in regulating biological functions as second messengers in cells [35]. Changes in intracellular states alter the production of ROS and the activation of apoptosis-inducing factors, causing apoptosis, which in turn amplifies the changes in intracellular redox states [36]. As expected, the results of this study showed that when porcine ESCs were exposed to ZEA, the levels of Ca^2+^ and ROS increased, which increased the expression of ERS-related genes and proteins, finally leading to cell apoptosis. compared with those in the control group, the levels of ERS marker genes and proteins decreased in response to 4-PBA decreased, the toxic damage was relieved. ERS is mainly caused by the perturbations that lead to the accumulation of misfolded proteins in the ER lumen and activate JNK. Meanwhile, studies have shown that ROS-induced DNA damage can trigger the activation of JNK, subsequently leading to cell death.

Classical MAPK is one of the downstream signal transduction pathways of ROS, which plays an important role in regulating gene expression, cell growth, and survival [37]. When ERS occurs, protein expression of CHOP is increased, which inhibits the expression of Bcl-2 and promotes apoptosis [31]. Bax and Bcl-2 are activated and inhibited, respectively, by p-JNK, representing JNK-induced apoptosis substrates [38]. JNK is located mainly in the cytoplasm. When cells are stimulated, JNK is phosphorylated and p-JNK enters the nucleus to exert its activity, leading to nuclear transcription factor (C-Jun) activation, which eventually activates pro-apoptotic gene expression, such as Bax [39,40]. This proves the results of this study, which showed that when ZEA induced ERS in ESCs, the phosphorylation level of JNK increased, and p-JNK translocation into the nucleus, resulting in increased expression of pro-apoptotic genes and proteins.

Bax is mostly located in the cytoplasm as a monomer. Bax can form oligomers that pass from the cytoplasm to the mitochondrial membrane and form polymers with Bcl-2 to increase mitochondrial permeability. In addition, Bax oligomers are introduced into the outer membrane to form channels that release Ca^2+^ and have a synergistic effect on Bax and activate the enzymatic cascade of Caspase 9 and caspase protease family, which eventually leads to apoptosis. Apoptosis can also be induced by continuous expression of JNK and CHOP, which block the cell cycle [41]. Our results showed that the JNK mRNA expression and the protein levels of JNK and p-JNK increased significantly in the ZEA group, as did the level of p-JNK in the nucleus. Bax expression increased significantly and Bcl-2 expression decreased significantly in response to ZEA, which increased Caspase 3 and Caspase 9 gene and protein levels, leading ESC apoptosis. The addition of 4-PBA alleviated the ERS and apoptosis induced by ZEA in ESCs. Previous studies have shown that the PERK-CHOP and ASK1-JNK signaling pathways are involved in the induction of apoptosis, which is consistent with the results of the present study [32,42].

## 4. Conclusions

The experimental results showed that ZEA induced a large amount of ROS and Ca^2+^ in ESCs, causing ERS, activating JNK signal pathway, which activated apoptosis genes, then reduced the cellular viability. Our findings clarified the toxic injury mechanism of ZEA on porcine ESCs and provided an effective theoretical basis for subsequent prevention and treatment research.

## 5. Materials and Methods

### 5.1. Chemical and Reagents

ZEA was obtained from Sigma Chemical Co. (St. Louis, MO, USA). Porcine ESCs were obtained from the cell bank of Wuhan Academy of Agricultural Sciences (Wuhan, China). Dulbecco’s modified Eagle’s medium (DMEM)/F12 was purchased from Hyclone (Logan, UT, USA). The SuperScript III kit and the SYBR green quantitative real-time PCR (qPCR) mix were obtained from Thermo Fisher Scientific (Waltham, MA, USA). 4-PBA was purchased from Target Molecule Corp. (Boston, MA, USA). The CCK-8 kit was obtained from Dojindo Laboratories (Kumamoto, Japan). The Cytotoxicity Assay Kit was obtained from Beyotime Biotechnology (Shanghai, China). The reactive oxygen species (ROS) kit, Cytotoxicity Assay Kit and Fluo-3-pentaacetoxymethyl ester (Fluo-3AM) were also obtained from Beyotime Biotechnology (Shanghai, China). The Dichloro-dihydro-fluorescein diacetate (DCFH-DA) assay kit was purchased from Nanjing SenBeiJia Biological Technology Co. (Nanjing, China). Annexin V-fluorescein isothiocyanate (FITC)/propidium iodide (PI) Apoptosis Detection Kit was obtained from Yeasen Biotech Co., Ltd. (Shanghai, China). The flow cytometer and laser confocal microscope were from Olympus (Tokyo, Japan). Bicinchoninic acid (BCA) Protein Assay Kit were purchased from Biosharp (Hefei, Anhui, China). Antibodies recognizing GRP 78, CHOP, ASK1, and PERK were purchased from Servicebio (Wuhan, China). The anti-JNK antibodies were from Proteintech Group, Inc. (Wuhan, China) and those recognizing phosphorylated (p)-JNK were procured from Cell Signaling Technology (Danvers, MA, USA). Antibodies recognizing β-actin, Bcl-2-associated X protein (Bax), B-cell lymphoma-2 (Bcl-2), Caspase 3, and Caspase 9 were obtained from ZEN-Bioscience (Chengdu, China). 

### 5.2. Cell Culture and Treatments

ESCs were cultivated in culture bottles (4 cm × 6 cm) in DMEM/F12 supplemented with 10% (*v/v*) fetal bovine serum (FBS), 100 U/mL penicillin, and 100 μg/mL streptomycin, and then cultured at 37 °C in a moistened incubator with 5% CO_2_. A 5 mg/mL stock solution of ZEA was prepared by dissolving 10 mg ZEA in 2 mL of dimethylsulfoxide. In this study, the 50% inhibitory concentration (IC50) of ZEA (15 μM) was determined according to pre-experiments and used for subsequent analyses.

To evaluate cell viability, ESCs were cultivated in 96-well plates at 1 × 10^5^ cells/mL for 24 h; and then cultured with 15 μM ZEA for 24 h. In some experiments, the cells were pretreated with 4-PBA for 3 h. The cell culture was the collected to detect cell viability using the CCK-8 assay (Kumamoto, Japan).

### 5.3. Detection of Cell Viability

ESCs at the logarithmic growth phase were inoculated in 96-well plates at about 8000 cells/well. After 24 h of culture, cells were treated with different concentrations of ZEA. After 24 h, 10 μL of CCK-8 reagent was added to each well and cultured for 2 h. Cell viability was calculated by determining the absorbance at 450 nm. The details of the method were reported by Cao et al. [43].

### 5.4. Assay for Cell Growth Status

ESCs in the logarithmic growth phase were sampled and their density was adjusted to 1 × 10^5^ per mL in a cell culture plate. After the cells adhered to the plate, the culture solution was changed to a cell culture medium containing different concentrations of ZEA and cultured for 4 h. The cells were then treated according to the instructions of Cytotoxicity Assay Kit (Shanghai, China), and then observed using laser confocal microscopy (Olympus, Tokyo, Japan). Under the microscope, the living cells showed green fluorescence and the dead cells showed red fluorescence.

### 5.5. Transmission Electron Microscopy (TEM) Analysis of Cell Morphology

Treated cells were collected by centrifugation and fixed for 4 h using 2.5% glutaraldehyde. The fixed cells were then dehydrated with ethanol, and transform to propylene oxide for 30 min, soaked with epoxy resin for 2.5 h, then at 40 °C and 60 °C for embedding and heating for 12 and 48 h, respectively. Cut into ultrathin sections, stained with lead citrate, and finally washed. The cell ultrastructure was detected using a high-resolution transmission electron microscope (TEOL-2010; Electronics Corporation, Tokyo, Japan) [22].

### 5.6. Detection of ROS Levels

After 24 h of treatment, the ESCs in each group were incubated with DCFH-DA in the dark for 30 min at 37 °C. The cells were collected by centrifugation and the fluorescence intensity of ROS were detected via fluorescence-activated cell sorting (FACS) Calibur flow cytometry (BD, Franklin Lakes, NJ, USA). For the details, see Cao et al. [43].

### 5.7. Detection of Ca^2+^ Levels

ESCs according to 8.0 × 10^4^/mL were exposed to ZEA in 6-well plate in for 24 h, after which the medium was removed, and the cells washed three times with Hank’s Balanced Salt Solution (HBSS). Then, each group was incubated with Fluo-3AM working solution in the dark for 30 min at 37 °C. The cells were collected by centrifugation and the fluorescence intensity of Ca^2+^ was detected via FACSC alibur flow cytometry.

### 5.8. Detection of the Apoptosis Rate

ESCs according to 8.0 × 10^4^/mL were exposed to ZEA in 6-well plate in for 24 h, harvested by centrifugation, and stained with Annexin V FITC/PI for 15 min in dark. The cell apoptosis rate was measured by flow cytometry using a previously published method [3,4].

### 5.9. qRT-PCR

Total RNA was extracted from ESCs using Trizol according to the methods reported by Cao et al. [43]. The total RNA was reverse transcribed to cDNA, which was quantified using qPCR according to previously published methods [4,22]. GAPDH (encoding glyceraldehyde-3-phosphate dehydrogenase) was detected as a housekeeping gene to normalize the mRNA levels. The primer sequences were synthesized by Sangon Biotech Co., Ltd. (Shanghai, China) and are described in Table 1.

### 5.10. Western Blotting Analysis

The cells were lysed using precooled radioimmunoprecipitation assay (RIPA) buffer at 4 ℃ for 20 min. After centrifugation, the protein concentration in the supernatant was detected using a bicinchoninic acid (BCA) protein kit (Hefei, Anhui, China). Then, protein levels were detected using western blotting (WB), which was carried out using the method detailed by Cao et al. [3,4].

### 5.11. Statistical Analysis

To calculate protein levels according to the average optical density (OD) of their bands on western blots, the Image J Analysis Software v1.8.0 (Media Cybernetics, Shanghai, China) was used. In the present study, data are presented as the mean ± SD (*n* = 3). Statistical analysis was performed using the Statistical Program for Social Sciences (SPSS) software version 20.0 (IBM Corporation, Armonk, NY, USA). Analysis of variance (ANOVA) was performed for comparisons among multiple groups. Histogram were drawn using the GraphPad Prism version 5.0 (GraphPad Inc., San Diego, CA, USA)

## Figures and Tables

**Figure 1 toxins-14-00758-f001:**
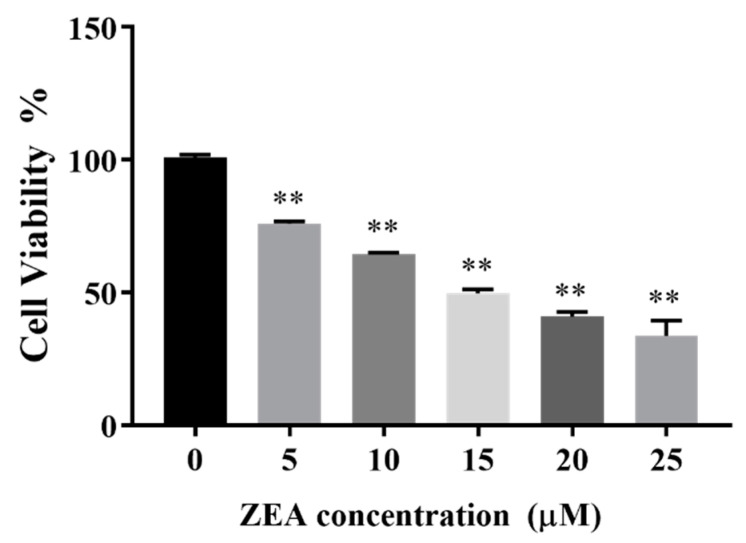
Effects of zearalenone (ZEA) on the viability in porcine endometrial stromal cells. ** indicates a significant difference compared with the control group (*p* < 0.01). ESCs were treated with different concentrations of ZEA. Treatment time: 24 h.

**Figure 2 toxins-14-00758-f002:**
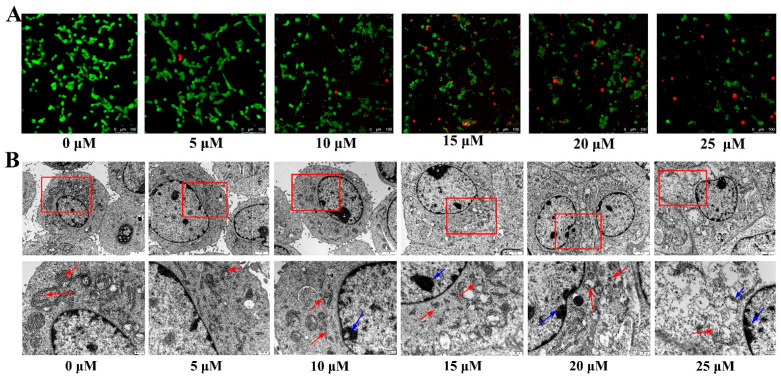
Effect of zearalenone (ZEA) on the growth state of porcine endometrial stromal cells. (**A**). Effect of ZEA on the growth status of cells (scale bar: 100 μm). Green fluorescence represents live cells and red fluorescence represents dead cells. (**B**). Effect of ZEA on cell ultrastructure (scale bar: 2 μm; 200 nm) The second set of images show magnified regions of the first set of images (red boxes). The red arrow represents the swelling and fracture of ER, the blue arrow represents nuclear chromatin margination. ESCs were treated with different concentrations of ZEA. Treatment time: 24 h.

**Figure 3 toxins-14-00758-f003:**
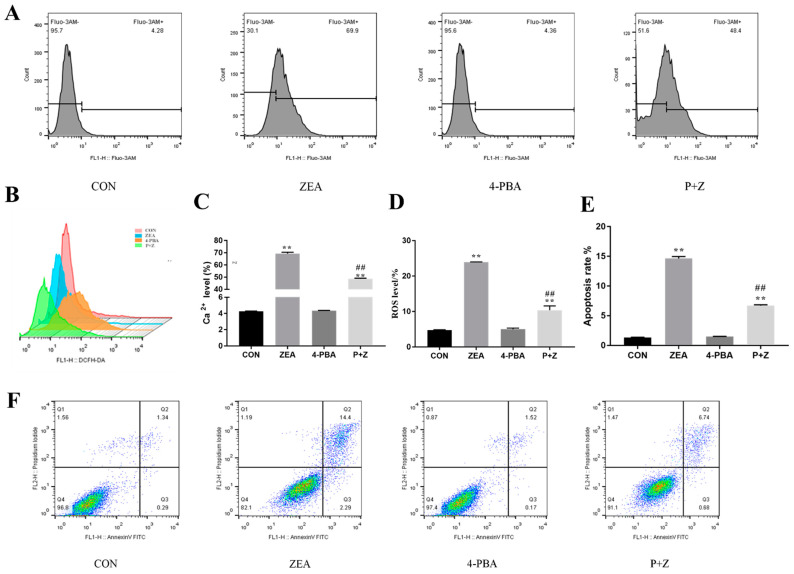
Effect of zearalenone (ZEA) on Ca^2+^, reactive oxygen species (ROS) levels and the apoptosis rate in porcine endometrial stromal cells. (**A**). The Ca^2+^ level detected by flow cytometry; (**B**). The ROS level detected by flow cytometry; (**C**). Effect of ZEA on intracellular Ca^2+^ level; (**D**). Effects of ZEA on intracellular ROS level; (**E**). Density maps of the apoptosis rate detected using flow cytometry; (**F**). Effect of ZEA on the apoptosis rate. CON, control; ZEA, cells treated with 15 μM ZEA; P + Z, cells treated with ZEA and 4-Phenylbutyrate (4-PBA). Treatment time: 24 h. “**” represents a highly significant difference compared with the control group. “##” indicates a highly significant difference compared with the ZEA group.

**Figure 4 toxins-14-00758-f004:**
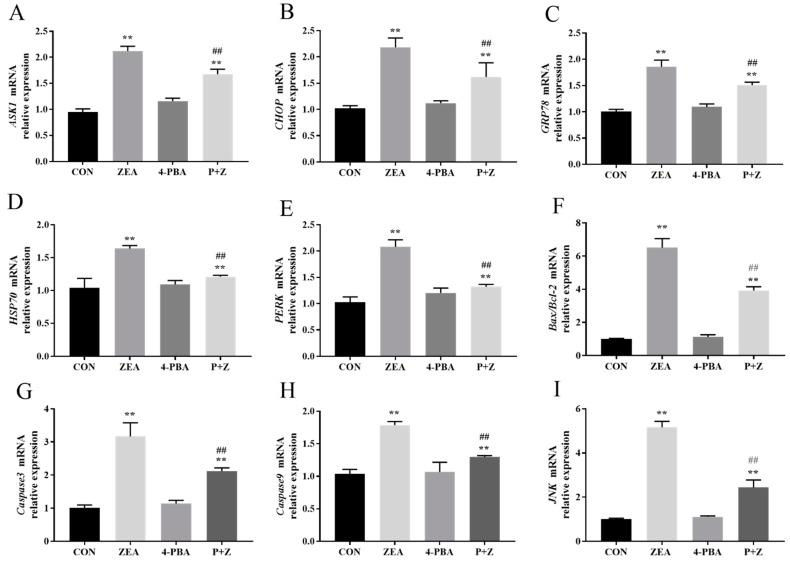
Effects of zearalenone (ZEA) on gene expression of porcine endometrial stromal cells (**A**) The effect of ZEA on ASK1 mRNA expression; (**B**) Effect of ZEA on CHOP mRNA expression; (**C**) Effect of ZEA on GRP78 mRNA expression; (**D**) Effect of ZEA on HSP70 mRNA expression; (**E**) Effect of ZEA on PERK mRNA expression; (**F**) Effect of ZEA on the Bax/BCL2 mRNA expression ratio; (**G**) Effect of ZEA on CASP3 mRNA expression; (**H**) Effect of ZEA on CASP9 mRNA expression; (**I**) Effect of ZEA on JNK mRNA expression. CON, control; ZEA, cells treated with 15 μM ZEA; P + Z, cells treated with ZEA and 4-Phenylbutyrate (4-PBA). Treatment time: 24 h. “**” represents a significant difference compared with the control group, “##” indicates a significant difference compared with the ZEA group.

**Figure 5 toxins-14-00758-f005:**
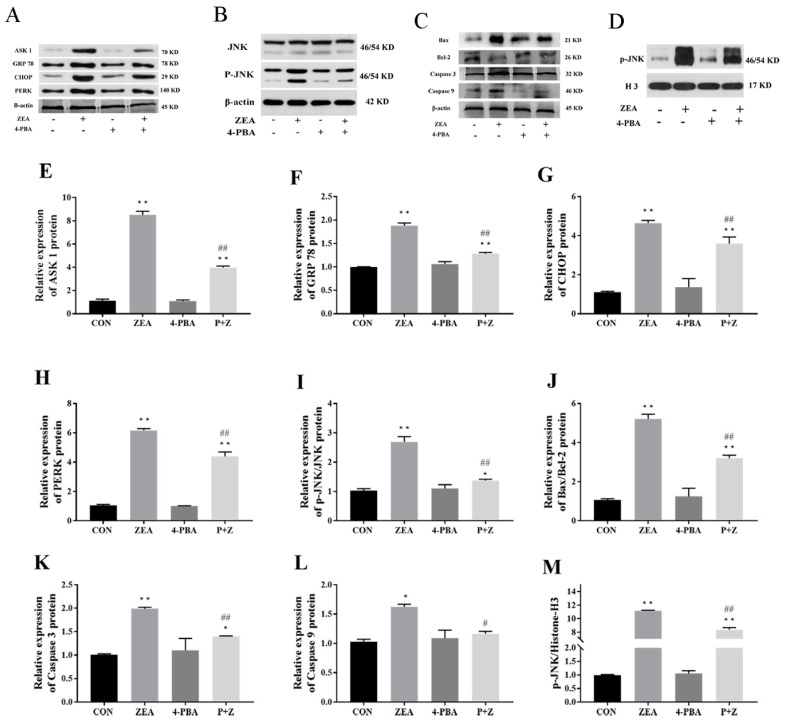
Effects of zearalenone (ZEA) on protein levels of ESCs. WB images of: (**A**). Levels of ERS related proteins in cells; (**B**). Levels of JNK and p-JNK protein in cells; (**C**). Levels of apoptosis-related protein in cells; (**D**). Levels of p-JNK in the nucleus. Histograms created from the WB data: (**E**). Levels of ASK1 in each group of cells; (**F**). Levels of GRP78 in each group of cells; (**G**). Levels of CHOP in each group of cells; (**H**). Levels of PERK in each group of cells; (**I**). The p-JNK/JNK ratio in each group of cells; (**J**). The intracellular Bax/Bcl-2 ratio in each group of cells; (**K**). Levels of Caspase 3 in each group of cells; (**L**). Levels of Caspase 9 in each group of cells; (**M**). Levels of p-JNK in the nucleus in each group of cells. CON, control; ZEA, cells treated with 15 μM ZEA; P + Z, cells treated with ZEA and 4-Phenylbutyrate (4-PBA). Treatment time: 24 h. “*” Represents a significant difference compared with the control group, “**” represents a highly significant difference compared with the control group, “#” indicates a significant difference compared with the ZEA group, “##” indicates a highly significant difference compared with the ZEA group.

**Table 1 toxins-14-00758-t001:** PCR primers used in the present study.

Gene	Accession Number	Primer	Primer Sequences (5′–3′)	Product Size/bp
*GAPDH*	XM_021091114.1	Forward	TGACCCCTTCATTGACCTCC	160
		Reverse	CCATTTGATGTTGGCGGGAT	
*GRP78*	X92446.1	Forward	GGCTCTACTCGCATCCCAAAG	115
		Reverse	CCTGAACAGCAGCACCGTAA	
*CHOP*	XM_005674378.2	Forward	CTTCACCACTCTTGACCCTG	170
		Reverse	CACTTTGTTTCCGTTTCCTG	
*HSP70*	NM_001123127.1	Forward	GCACGAGGAAAGCCTTAGAG	166
		Reverse	GGAGAAGATGGGACGACAAA	
*PERK*	XM_003124925.4	Forward	TCTTGGTAGGGTCTGATGAA	132
		Reverse	GCTTGTAGTATGGCAGGTAAT	
*BCL2*	XM_021099602.1	Forward	TCCAGGCAGTTTAATACATTC	80
		Reverse	TCCCTTTATACACTGGGTGA	
*Bax*	XM_003127290.5	Forward	TGGAGCAGGTGCCTCAGGAT	171
		Reverse	TGCCGTCAGCAAACATTTCG	
*CASP3*	NM_214131.1	Forward	TCTAACTGGCAAACCCAAAC	85
		Reverse	AGTCCCACTGTCCGTCTCAA	
*CASP9*	XM_013998997.2	Forward	ACAGGACCGCCGACAGTAAC	154
		Reverse	TCCCTCCAGGAGACAAACCC	
*JNK*	XM_021073087.1	Forward	TCAGGCAAGGGATTTGTTAT	141
		Reverse	TCAGGTATCTTTGGTGGTGG	

## Data Availability

Not applicable.

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
