# Peer review of "Zearalenone Induces Apoptosis in Porcine Endometrial Stromal Cells through JNK Signaling Pathway Based on Endoplasmic Reticulum Stress"

_toxins, 2022, doi:10.3390/toxins14110758_

Round 1
Reviewer 1 Report
The authors have studied the effect of zearalenone in porcine ES cells at the molecular level. They have evaluated a) cell viability and structure after 4 or 24h of ZEA treatment at different concentrations; b) Calcium, ROS and apoptosis after 24h of a single concentration of ZEA; c) mRNA gene expression and protein level of 9 biomarkers of endoplasmic reticulum stress or apoptosis pathways.
I suggest the following modifications
Introduction:
In the first paragraph summarising ZEA toxic effects, two citations (3 and 4) are self-citations which should be avoided. On the contrary some EFSA scientific opinions could be provided (EFSA Journal 2011; 9(6):2197; EFSA Journal 2017; 15(7):4851, 123 pp. doi:10.2903/j.efsa.2011.2197. https://doi.org/10.2903/j.efsa.2017.4851). These scientific opinions are vey good reviews of the subject.
Two paragraphs are dedicated to describe the molecular pathways without any relationship with the mycotoxin or its known toxic effects. Moreover, without a scheme it is difficult to understand.
In summary, a more focused introduction is needed.
Page 2. Line 55, suppress Lee et al and…
The final aim of this piece of work is to prevent and treat mycotoxicosis but I do not see this objective accomplished. It should be eliminated from the introduction and conclusion. Otherwise, more explanation is necessary.
Results
Treatment conditions should be specified in the figure legend and/or in the text.
Fig 1 and 2. Treatment time is 24h?
Fig. 3, 4, 5 Only at IC50? Treatment time?
How does the ERS inhibitor acts?
Discussion
It sould be re-written. It is not ordered and the results are not really discussed or compared with others. Instead, a description of the molecular pathways is repeated but the role the ZEA may have is not really discussed. I find a diagram necessary.
Conclusion
See previous comment.
Author Response
Point 1: Introduction: In the first paragraph summarising ZEA toxic effects, two citations (3 and 4) are self-citations which should be avoided. On the contrary some EFSA scientific opinions could be provided (EFSA Journal 2011; 9(6):2197; EFSA Journal 2017; 15(7):4851, 123 pp. doi:10.2903/j.efsa.2011.2197. https://doi.org/10.2903/j.efsa.2017.4851). These scientific opinions are vey good reviews of the subject.
Two paragraphs are dedicated to describe the molecular pathways without any relationship with the mycotoxin or its known toxic effects. Moreover, without a scheme it is difficult to understand.
In summary, a more focused introduction is needed.
Response 1: Thank you for your valuable suggestions. We read two papers recommended by the reviewer and revised them in the corresponding part of the introduction. We have modified the next two paragraphs to simplify the presentation and show the purpose of this study.
Point 2: Page 2. Line 55, suppress Lee et al and…
The final aim of this piece of work is to prevent and treat mycotoxicosis but I do not see this objective accomplished. It should be eliminated from the introduction and conclusion. Otherwise, more explanation is necessary.
Response 2: Thank you for your valuable comment. We have removed the redundant reference representation after Lee et al.
In addition, the aim of this study was modified in the introduction and conclusion, which clarified the molecular mechanism of ZEA-induced toxic injury.
Point 3: Results
Treatment conditions should be specified in the figure legend and/or in the text.
Fig 1 and 2. Treatment time is 24h?
Fig. 3, 4, 5 Only at IC50? Treatment time?
How does the ERS inhibitor acts?
Response 3: Thank you for your careful review. In Fig. 1 and 2, the treatment time of ZEA is 24 h. In Fig. 3, 4, and 5, only IC50 was selected and the treatment time is 24 h. We have been added in the figure legend. ERS inhibitor, pretreatment for 3 h followed by 15 μM ZEA for 24 h. Details description in 5.2 Cell Culture and Treatments.
Point 4: Discussion
It sould be re-written. It is not ordered and the results are not really discussed or compared with others. Instead, a description of the molecular pathways is repeated but the role the ZEA may have is not really discussed. I find a diagram necessary.
Response 4: Thank you for your careful review. We have made some modifications in the discussion and added a graphical abstract at the end of the manuscript.
Point 5: Conclusion
See previous commen
Response 5: Thank you for your valuable comment and important point. We have made modifications according to the previous comments.
Reviewer 2 Report
The study investigated the effects of zearalenone on the activation of JNK signaling pathway in porcine endometrial stromal cells.
The article is well written and in a coherent manner. Its structure is adequate.
The results are presented in detail. The conclusions are supported by results.
Author Response
Thank you for your review and affirmation of the content of this article.
Reviewer 3 Report
The study is original and very interesting, involving a huge amount of work. Congratulations.
I suggest just very few minor corrections.
1. In Conclusions section, instead of first sentence or after it, tell something more about your own results
2. line 55, you have cited one reference as number, others by names. Put also numbers of references
3. in References section, as I know, the Instructions for authors ask to add doi code of the References
Author Response
Point 1: In Conclusions section, instead of first sentence or after it, tell something more about your own results
Response 1: Thank you very much for your comment. We have revised the conclusion according to the comment.
Point 2: line 55, you have cited one reference as number, others by names. Put also numbers of references
Response 2: Thank you for your valuable and important review. According to your valuable suggestions, we have revised in the manuscript.
Point 3: in References section, as I know, the Instructions for authors ask to add doi code of the References
Response 3: Thank you for your careful review. We have added doi code in references.